# Layer-Wise Coordination between Encoder and Decoder for Neural Machine Translation

**Tianyu He**[1][†][*]
hetianyu@mail.ustc.edu.cn

**Xu Tan**[2][†]
xuta@microsoft.com

**Yingce Xia**[2]
yingce.xia@microsoft.com

**Di He**[3]
di_he@pku.edu.cn

**Tao Qin**[2]
taoqin@microsoft.com

**Zhibo Chen**[1]
chenzhibo@ustc.edu.cn

**Tie-Yan Liu**[2]
tie-yan.liu@microsoft.com

[1]CAS Key Laboratory of Technology in
Geo-spatial Information Processing and Application System,
University of Science and Technology of China
[2]Microsoft Research
[3]Key Laboratory of Machine Perception, MOE, School of EECS, Peking University

## Abstract

Neural Machine Translation (NMT) has achieved remarkable progress with the quick evolvement of model structures. In this paper, we propose the concept of layer-wise coordination for NMT, which explicitly coordinates the learning of hidden representations of the encoder and decoder together layer by layer, gradually from low level to high level. Specifically, we design a layer-wise attention and mixed attention mechanism, and further share the parameters of each layer between the encoder and decoder to regularize and coordinate the learning. Experiments show that combined with the state-of-the-art Transformer model, layer-wise coordination achieves improvements on three IWSLT and two WMT translation tasks. More specifically, our method achieves 34.43 and 29.01 BLEU score on WMT16 English-Romanian and WMT14 English-German tasks, outperforming the Transformer baseline.

## 1   Introduction

Neural Machine Translation (NMT) is a challenging task that attracts a lot of attention in recent years [5, 2, 18, 29, 7, 33, 28, 27, 35, 37, 9], and the structure of NMT models has evolved quickly. The first design of NMT model is based on Recurrent Neural Networks (RNNs) [5]. Then the attention mechanism [2] is introduced to better model the alignment between source and target tokens. Deeper architectures are adopted later to increase the expressiveness of NMT models [36, 7, 33]. Recently, Convolutional Neural Network [7] and self-attention [33] based models are invented, which achieve the state-of-the-art performance in many broadly adopted translation tasks.

While those models employ different basic building blocks (e.g., RNN, CNN, or self-attention), they are all under the typical encoder-decoder framework: The encoder takes the source tokens as inputs and generates a set of hidden representations for those tokens layer by layer, gradually from

---

[*]The work was done when the first author was an intern at Microsoft Research Asia.
[†]The first and second author contribute equally to this work.

low level to high level. Then the decoder takes the last-layer (the highest level) representations from the encoder as inputs, generates hidden representations for each target position from low-level layers to high-level ones, and finally generates a token based on the last-layer representations. We can see that the generation of hidden representations for target tokens, no matter high level or low level, are all based on the highest-level representations of the source sentence. Our case study and attention visualization (Section 5.3) show that attending the low layer of the decoder to the high-level representations of the encoder causes unfocused attention and harms translation quality.

Then questions come out naturally: Why should the low-level representation of a target token base on the highest-level ones of source tokens? Why not attending each layer of the decoder to each corresponding layer of the encoder? These questions exactly motivate our work.

In this paper, we propose to coordinate the learning of the encoder and decoder of an NMT model layer by layer. The encoder and decoder of our model have the same number of layers, and the $i$-th layer in the decoder is aligned and coordinated with the $i$-th layer of the encoder. The hidden representation of a source token in the $i$-th layer is generated from the hidden representations of all source tokens in the ($i$-1)-th layer using the self-attention mechanism, and that of a target token in the $i$-th layer is generated from the hidden representations of all source tokens and preceding target tokens in the ($i$-1)-th layer using a mixed attention mechanism. To further coordinate the learning between the encoder and decoder, we share parameters of the encoder and decoder. This new model has several advantages compared with existing models.

First, through layer-wise coordination, the information from the source and target sentence will meet earlier, starting from the low-level representations. Consequently, the decoder can leverage more fine-grained source information when generating target tokens, instead of only using high-level representations outputted by the encoder in the previous model structure. Such an approach has been shown to be effective for other NLP tasks such as text matching [16, 10, 19] or non-autoregressive machine translation [8].

Second, through layer-wise coordination and parameter sharing, we ensure that the hidden representations in two corresponding layers of the encoder and decoder are in the same (or closely related) semantic level. Note that even if existing models can have the same number of layers in their encoder and decoder, there is no correspondence between an encoder layer and a decoder layer since their parameters are freely learned. Furthermore, parameter sharing allows us to stack more layers under the constraint of model size, without loss of model capacity but regularizing training process.

The idea of layer-wise coordination can be applied to most existing model architectures, including RNN [5, 2, 18, 29], CNN [7] and Transformer [33]. In this work we apply layer-wise coordination to Transformer, considering its super accuracy on several benchmark tasks. Experiments show that our method outperforms strong baselines on three IWSLT tasks and two WMT tasks. In particular, we achieve 34.43 and 29.01 BLEU score on WMT16 English-Romanian and WMT14 English-German tasks.

## 2  Background

### 2.1  Encoder-Decoder Framework

Given a bilingual sentence pair $(x, y)$, an NMT model learns its parameter $\theta$ by maximizing the log-likelihood $P(y|x;\theta)$, which is usually decomposed into the product of the conditional probability of each target word: $P(y|x;\theta) = \prod_{t=1}^{m} P(y_t|y_{<t}, x; \theta)$, where $m$ is the length of sentence $y$, $y_{<t}$ are the target tokens before position $t$.

An encoder-decoder framework [5, 2, 18, 29, 7, 33] is usually adopted to model the conditional probability $P(y|x;\theta)$. The encoder maps the input sentence $x$ into a set of hidden representations $h$, and the decoder generates the target token $y_t$ at position $t$ using the previously generated target tokens $y_{<t}$ and the source representations $h$. Both the encoder and decoder can be implemented by different structure of neural models, such as RNN (LSTM/GRU) [5, 2, 18, 29], CNN [7] and self-attention [33]. Besides the basic component of the encoder and decoder, a source-target attention mechanism [2] is usually adopted to selectively focus on the source representations when generating a target token.

Different from the typical encoder-decoder framework, our layer-wise coordination allows each layer of the decoder to directly leverage the hidden representations from the corresponding layer (instead of the last layer) of the encoder.

## 2.2 Self-attention based Network

Self-attention has been used in many previous works [4, 21, 22, 33, 15]. [33] first introduces self-attention into Neural Machine Translation. For a single self-attention layer, it utilizes a cross-position self-attention to extract information from the tokens in the whole sentence, and then a position-wise feed-forward network to increase the non-linearity. The self-attention is formulated as:

$$\text{Attention}(Q, K, V) = \text{softmax}(\frac{QK^T}{\sqrt{d_{\text{model}}}})V, \tag{1}$$

where $d_{\text{model}}$ is the dimension of hidden representations. The embedding size, the input and output size of self-attention are all set as $d_{\text{model}}$. For the self-attention inside the encoder layer, $Q, K, V \in \mathbb{R}^{n * d_{\text{model}}}$, while for the self-attention inside the decoder layer, $Q, K, V \in \mathbb{R}^{m * d_{\text{model}}}$, where $n$ and $m$ is the length of source and target sentence. For the attention cross the encoder and decoder from the source to the target (i.e., source-target attention), $Q \in \mathbb{R}^{m * d}$, $K, V \in \mathbb{R}^{n * d_{\text{model}}}$. All the $Q, K, V$ come from the hidden representations of the corresponding encoder/decoder layer, but projected by different parameter matrices: $W_Q, W_K$ and $W_V$. The position-wise feed-forward network consists of a two-layer linear transformation with ReLU activation in between:

$$\text{FFN}(x) = \max(0, xW_1 + b_1)W_2 + b_2. \tag{2}$$

The feed-forward network is applied on every layer of both the source and target sentence.

# 3 Layer-wise Coordination

In this section, we present the idea of layer-wise coordination. In principle, layer-wise coordination can be applied to any encoder-decoder based models, including RNN, CNN and Transformer. In this work, we directly focus on Transformer considering that it achieves very good accuracy on multiple translation tasks.

Layer-wise coordination modifies the structure of Transformer from two aspects: First, each layer in the decoder attends to the corresponding layer in the encoder. That is, the encoder and decoder have the same number of layers and layer $i$ in the decoder can extract information directly from layer $i$ in the encoder, instead of the last layer of the encoder like Transformer. While the decoder of Transformer uses a separate encoder-decoder attention module to extract information from the source sentence and a self-attention module to extract information from previous target tokens, we merge the two attentions into one, which is called as mixed attention, to coordinate the learning between source and target. Second, we share the parameters of attention and feed-forward layer between the encoder and decoder, in order to ensure the outputs of the corresponding layers of the encoder and decoder are in the same (or closely related) semantic level, and thus enhance layer-wise coordination.

The overall structure of our model with layer-wise coordination is shown in Figure 1. The source and target sentences are concatenated and processed by the model layer by layer coordinately. We stack $N$ layers and $N$ can be varied according to different sizes of datasets. We introduce the key components of the model as follows.

**Mixed Attention** In order to coordinate the learning of source and target tokens in each layer, the decoder of our model uses a mixed attention for a target token to extract cross-position information from both the source and preceding target tokens. The attention mechanism is shown as the "Mixed Attention" in Figure 1.

To enable the above attention mechanism, we add an extra mask on the dot product of $Q$ and $K$ based on Equation 1 to prevent attending to the illegal positions (i.e., future target tokens):

$$\text{Mixed\_Attention}(Q, K, V) = \text{softmax}(\frac{QK^T}{\sqrt{d_{\text{model}}}} + M)V,$$
$$M(i, j) = \begin{cases} 0, \ j < n \lor j \leq i + n \\ -\infty, \ \text{otherwise} \end{cases}, \tag{3}$$

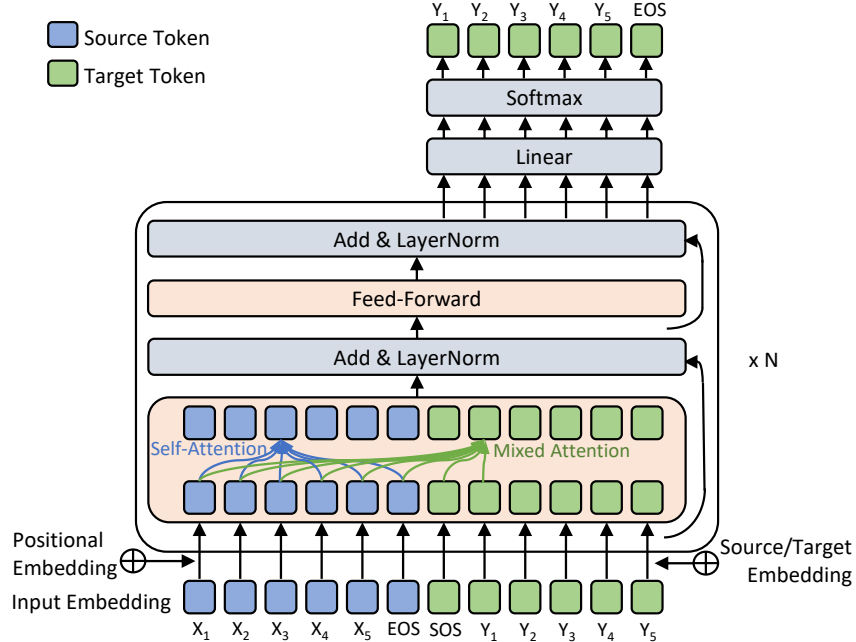

Figure 1: Our proposed layer-wise coordination model for neural machine translation.

where $Q \in \mathbb{R}^{m*d_{\text{model}}}$, $K, V \in \mathbb{R}^{(n+m)*d_{\text{model}}}$, and $M \in \mathbb{R}^{m*(n+m)}$ is a mask matrix, with $n$ and $m$ being the length of source and target sentence. When $M(i, j)$ equals to $-\infty$, the corresponding position in softmax output will approach zero, which prevents position $i$ from attending to position $j$.

**Position Embedding**    Since self-attention has no recurrent operation like RNN or convolution like CNN, we need explicitly inject some information to indicate the absolute or relative position of a word to the model. In order to keep the original order of the concatenated sentence, we use the resettable position embedding. The position of source tokens starts with zero, and for the target tokens, instead of increasing the position upon the end of source sentence, we reset the position from zero again. As the position embedding function, we follow [8, 33] and use the sine and cosine functions to format the embedding vectors: $p(pos, k) = sin(pos/10000^{k/d_{\text{model}}})$ (if $k$ is even) or $cos(pos/10000^{k/d_{\text{model}}})$ (if $k$ is odd), where $pos$ is the position and $k$ is the index of the hidden dimension.

**Source/Target Embedding**    Since the source and target tokens share the same model parameters, we need to give the model a sense of which language a token comes from it is receiving. The resettable position embedding alone cannot identify the language of a word token. We introduce two embeddings which represent the source and target language respectively. Every position of the source and target tokens is added with the corresponding source/target embedding, which are learned end-to-end during the model training process. The source/target embeddings are demonstrated to be extremely important to train the model in our experiments.

## 4    Experimental Setup

### 4.1    Datasets

We evaluate our model on several widely used translation tasks, including IWSLT14 German-English (briefly, De-En), IWSLT14 Romanian-English (briefly, Ro-En), IWSLT14 Spanish-English (briefly, Es-En), WMT16 English-Romanian (briefly, En-Ro) and WMT14 English-German (briefly, En-De).

**IWSLT14 German/Romanian/Spanish-English (De-En/Ro-En/Es-En)**    We use the datasets extracted from IWSLT 2014 evaluation campaign [3] [3], which consist of 153K/182K/181K training

sentence pairs for De-En/Ro-En/Es-En. For Ro-En/Es-En, we concatenate dev2010, tst2010, tst2011 and tst2012 as the validation set and use tst2014 as the test set. For De-En, we use 7K data split from the training set as the validation set and use the concatenation of dev2010, tst2010, tst2011 and tst2012 as the test set, which is widely used in prior works [23, 1, 11]. We also lowercase [4] the sentences of De-En following the common practice. Sentences are encoded using sub-word types based on byte-pair-encoding (BPE) [25] [5], which has a shared vocabulary of about 31K/39K/34K sub-word tokens for De-En/Ro-En/Es-En.

**WMT16 English-Romanian (En-Ro)**  We use the same dataset and pre-processing techniques as [24], which result in 2.8M sentence pairs for training. We use the concatenation of newstest2013 and newstest2014 as the validation set and newstest2016 as the test set [24]. Sentences are also encoded using BPE with a shared vocabulary of 36K sub-word tokens.

**WMT14 English-German (En-De)**  We use the same dataset as [17], which comprises 4.5M sentence pairs for training. We use the concatenation of newstest2012 and newstest2013 as the validation set and newstest2014 as the test set [6]. Sentences are also encoded using BPE with a shared vocabulary of 40K sub-word tokens.

## 4.2  Model Configurations

For small datasets De-En/Ro-En/Es-En, we choose the *small* configuration with the model hidden size $d_{\mathrm{model}} = 256$ and feed-forward hidden size $d_{\mathrm{ff}} = 1024$. For relative larger datasets En-Ro and En-De, we choose the *big* configuration with $d_{\mathrm{model}} = 1024$ and $d_{\mathrm{ff}} = 2048$. We used the same number of heads as Transformer (4/8/16 for *small/base/big* configuration). For fair comparison with Transformer, we perform our experiments under the constraint that the number of parameters is similar with Transformer. Since our model structure shares the parameters between the encoder and decoder, we can stack more layers under the same parameter constraint. We stack 14 layers for both *small* and *big* configurations, which have roughly the same number of parameters with the 6-layer *transformer_small* and *transformer_big* counterparts [32].

## 4.3  Training and Inference

During training, we concatenate the source and target sentence together and batch the concatenated sentences with approximate sentence lengths with zero padded at the end of each sentence to ensure exactly the same length in one mini-batch. Each mini-batch on one GPU contains roughly 4096 tokens. We train our models for En-Ro and En-De with 8 NVIDIA Tesla M40 GPUs on one machine. We only use one M40 GPU to train the model for De-En/Ro-En/Es-En tasks as it is of both small model size and data size. The validation sets in all the datasets are used for hyper-parameter tuning and early stopping. We choose the Adam optimizer [13] with $\beta_1 = 0.9$, $\beta_2 = 0.98$, $\varepsilon = 10^{-9}$ and use the learning rate schedule in [33].

During inference, we generate the target token autoregressively, regarding the source tokens as the previously generated tokens. We append the source tokens with the end-of-sentence (EOS) token, and then with the start-of-sentence (SOS) token, and feed them into the model to generate the first target token. We decode with beam search and set beam width $beam = 6$ and length penalty $\alpha = 1.1$ on all datasets except for En-De, where we use $beam = 4$ and $\alpha = 0.6$ to be consistent with [33]. We evaluate the translation quality by tokenized case-sensitive BLEU [20] with multi-bleu.pl[7], except for De-En where we use case-insensitive BLEU to follow the common practice and En-Ro where we use detokenized BLEU to be consistent with [24, 7] for comparison. Larger BLEU means better translation quality.

## 5 Results

### 5.1 Compared with Previous NMT Models

We evaluate our proposed model on the five translation tasks and compare with the previous models that are under the typical encoder-decoder framework. First, we compare with the Transformer baseline, which is trained with the tensor2tensor codes [32]. We adopt the *small* configuration for our model on IWSLT14 De-En/Ro-En/Es-En tasks, and *big* configuration on WMT16 En-Ro and WMT14 En-De tasks. On WMT14 En-De task, we also run an extra *base* configuration for our model to be comparable with the *transformer_base* model available in the Transformer paper, where the $d_{\mathrm{model}}$ and $d_{\mathrm{ff}}$ are set to 512 and 2048. On this task, we reproduce the BLEU score in the Transformer paper [33] on *transformer_big* and *transformer_base* configurations[8] and therefore we just list the original number in the paper. Second, we also compare with the results of other RNN/CNN-based models reported in the previous works. The BLEU scores are listed in Table 1 and 2.

On IWSLT14 De-En task, our method achieves 35.07 in terms of BLEU score, with 2.21 points improvement over the Transformer baseline. We also compare with some RNN-based models and our model achieves great improvements. On IWSLT14 Ro-En/Es-En task, we also surpass the Transformer baseline for 1.08/1.93 BLEU score.

On WMT16 En-Ro task, our model achieves 1.73 BLEU score improvements over the Transformer baseline. Compared with the RNN-based [24] and CNN-based [7] encoder-decoder models, our method also outperforms both of these models.

On WMT14 En-De task, we compare with Transformer in both the *base* and *big* model configurations. We achieve 1.03 BLEU score improvement over the *transformer_base* model and advance the *transformer_big* model with a new state-of-the-art BLEU score of 29.01. Again, we outperform all the RNN/CNN-based encoder-decoder framework in terms of BLEU score.

| Task | Method | BLEU |
|---|---|---|
| De-En | MIXER [23] | 21.83 |
| | AC+LL [1] | 28.53 |
| | NPMT [11] | 28.96 |
| | Dual Transfer Learning [34] | 32.35 |
| | Transformer (small) | 32.86 |
| | Our method (small) | **35.07** |
| Ro-En | Transformer(small) | 29.64 |
| | Our method (small) | **30.72** |
| Es-En | UEDIN[3] | 37.29 |
| | Transformer(small) | 38.57 |
| | Our method (small) | **40.50** |

Table 1: BLEU scores on IWSLT 2014 translation tasks compared with transformer baseline and other RNN/CNN-based models.

| Task | Method | BLEU |
|---|---|---|
| En-Ro | GRU[24] | 28.10 |
| | ConvS2S[7] | 30.02 |
| | Transformer (big) | 32.70 |
| | Our method (big) | **34.43** |
| En-De | ByteNet [12] | 23.75 |
| | GNMT+RL [36] | 24.60 |
| | ConvS2S [7] | 25.16 |
| | MoE [26] | 26.03 |
| | Transformer (base) [33] | 27.30 |
| | Transformer (big) [33] | 28.40 |
| | Our method (base) | 28.33 |
| | Our method (big) | **29.01** |

Table 2: BLEU scores on WMT translation tasks compared with transformer baseline and other RNN/CNN-based models.

### 5.2 Model Variations

**Ablation Study**   To evaluate the importance of different components of layer-wise coordination model, we mask each component of our model and test the performance changes on De-En task. We follow the same decoding strategy as described in Section 4.3 and compare the BLEU scores changes on the test set, as listed in Table 3.

The first row in Table 3 is the basic parameter setting for our model. First, we do not use weight sharing in the second row, but for fair comparison we reduce half of the model layers to ensure the same amount of model parameters. We can see sharing weight indeed outperforms the non-

|  | #parameter | BLEU | Δ |
|---|---|---|---|
| Our model | 19.07M | 35.07 | |
| Our model w/o weight sharing | 19.07M | 33.96 | 1.11 ↓ |
| Our model w/o mixed attention | 19.07M | 33.77 | 1.30 ↓ |
| Our model w/o source/target embedding | 19.07M | 32.80 | 2.27 ↓ |
| Our model w/o position embedding | 19.07M | 18.46 | 16.61 ↓ |

Table 3: Ablation study on our proposed model on De-En task.

| #layer | 10 | 14 | 18 | 22 | #layer | 4 | 6 | 8 | 10 |
|---|---|---|---|---|---|---|---|---|---|
| Our method | 34.32 | 35.07 | 35.31 | 35.05 | Baseline | 32.78 | 32.86 | 32.72 | 32.67 |

Table 4: The BLEU scores under different number of layers for our method and the baseline on De-En task.

| Source (De) | zwei minuten später passierten drei dinge gleichzeitig. |
|---|---|
| Reference (En) | two minutes later, three things happened at the same time. |
| Transformer | two minutes later, three things happened. |
| Our model | two minutes later, three things happened at the same time. |
| Source (De) | mit 17 wurde sie die zweite frau eines mandarin, dessen mutter sie schlug. |
| Reference (En) | at 17 she became the second wife of a mandarin whose mother beat her. |
| Transformer | at the age of 17, she turned into a mandarin second woman whose mother beat her. |
| Our model | at 17, she became the second woman of a mandarin whose mother beat her. |
| Source (De) | und ich erwiderte: "wie kommuniziert ihr denn nun?" |
| Reference (En) | and i said, "well, how do you actually communicate?" |
| Transformer | and i said, "how does you communicates?" |
| Our model | and i said, "how do you communicate?" |

Table 5: Translation examples on De-En dataset from our model and the Transformer baseline.

sharing counterpart. Second, if we separate the mixed attention into self-attention and source-target attention as usual, the BLEU score also drops. Third, removing position embedding and source/target embedding both hurt the model performance, especially for removing position embedding. The aforementioned results demonstrate the importance of each component of our model.

**Varying the Number of Layers** We vary different number of layers to investigate how our model performs. Table 4 shows the BLEU score on De-En task with 10/14/18/22 layers. For a fair comparison, we also vary the number of layers of baseline model with 4/6/8/10 to ensure the similar number of parameters between two models (i.e., our model with 10 layers has similar number of the baseline model with 4 layers). We can observe that the BLEU scores do not nearly change or even drop when increasing the number of layer for baseline model, may due to overfitting. However, our layer-wise coordination can build a deep model up to 18 layers on this task, achieving a record-breaking 35.31 BLEU score. When increasing to 22 layers, which is an extremely deep configuration for NMT, our model also drops to 35.05. We will investigate deeper model training through our layer-wise coordination for future work.

## 5.3 Case Study

**Case Analysis** Table 5 shows several translation examples produced by our model compared with Transformer baseline. We can see that our layer-wise coordination model generates more adequate, fluent and accurate sentences. For the first case, the Transformer suffers from the adequacy problem that misses the information "at the same time" while our model catches this information accurately. In the second case, although Transformer nearly translates the meaning of the source sentence adequately, the result suffers from fluency compared to our model. In the third case, Transformer generates the sentence mistakenly with third-person singular while our model handles this case well.

**Attention Visualization** In order to give a deep understanding why our model works better, we visualize the attention weights of our model and Transformer for the first case from Table 5. In this case, we analyze why Transformer misses the information "at the same time" while our model

translates it successfully. We investigate what information the model attends on when generating the next token after "happened". The attention weights are from the first layer of the decoder for both models. Transformer, as the typical encoder-decoder framework, uses two attentions to extract information from source and target separately and it extracts the source information from the last layer of hidden representations of the encoder, and thus the low layer of the decoder may not extract this high-level representation precisely. As shown in Figure 2, when generating the next token of "happened", it attends to diverse tokens, such as "passierten", "gleichzeitig","." and "EOS", which cause the generation of "." and finish the translation earlier. The target self-attention in Figure 3 alone cannot provide much information for the correct prediction. However, in our mixed attention as shown in Figure 4, we can observe that the attention weights mostly focus on the source token "gleichzeitig" that means "simultaneously" in English, previous generated token "happened" as well as the current position, which can precisely help the model generate the next token "at" for a beginning of the phrase "at the same time". More cases can be found in the supplementary materials (part A). This kind of cases show the advantages of our layer-wise coordination learning over the typical encoder-decoder based models.

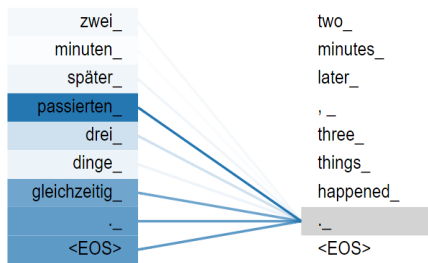

Figure 2: Source to target attention in Transformer.

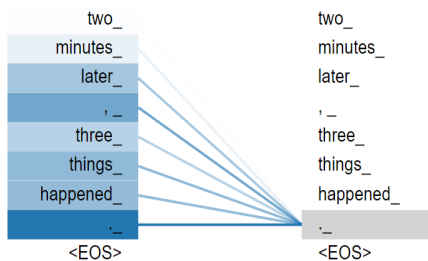

Figure 3: Target self-attention in Transformer.

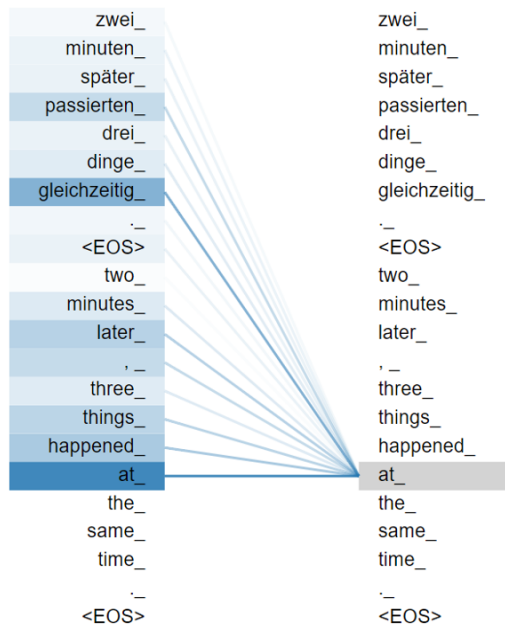

Figure 4: Mixed attention in our model.

### 5.4 Discussions on Mixed Attention

In NMT, the generation of the target word depends on both source and target contexts, where source contexts affect the adequacy of the generated sentence while target contexts have impact on the fluency [6, 14, 30, 31]. Our mixed attention is designed to better coordinate the learning of encoder and decoder by extracting cross-position information from both the source and preceding target tokens in one and the same attention function. In this way, the model automatically learns the preference on the source or target contexts when generating the target token, which will be beneficial when tackling with the adequacy and fluency problem, as a by-product of our model design.

[30] also developed a context gate on RNN-based model to trade off the context information from source and target. Here we compare our proposed model with [30] on De-En task by implementing it with Transformer, since it is originally implemented for RNN-based NMT model. The implementation details can be found in the supplementary materials (part B). For fair comparison, we just use layer-wise coordination without weight sharing between the encoder and decoder (the corresponding BLEU score is 33.96 and parameter size is 19.07M as shown in Table 3). Our implemented Transformer version of [30] has a 6-layer encoder and decoder with parameter size of 19.08M. The BLEU score is 33.02, with 0.94 point lower than our method.

We also show more visualization cases on our mixed attention in the supplementary materials (part C).

## 6 Conclusion

In this work, we improved existing NMT models through layer-wise coordination of the encoder and decoder. Our method aligns the $i$-th layer of the encoder to the $i$-th layer of the decoder and coordinates the learning of the hidden representations of source and target sentences layer by layer, by sharing the parameters of the aligned layers. Experiments on several translation tasks demonstrated our proposed model outperforms the Transformer baseline as well as other RNN/CNN-based encoder-decoder models.

For future works, we will apply the idea of layer-wise coordination to other sequence to sequence tasks, such as question answering and image captioning. We will also investigate better ways to coordinate the learning and interaction between the source and target. Furthermore, we will study how to leverage layer-wise coordination to train deeper NMT models.

## 7 Acknowledgement

This work was partially supported by the National Key Research and Development Program of China under Grant No. 2016YFC0801001, the National Program on Key Basic Research Projects (973 Program) under Grant 2015CB351803, NSFC under Grant 61571413, 61632001, 61390514. We thank all the anonymous reviewers for their valuable comments on our paper.

## Footnotes

[3]https://wit3.fbk.eu/mt.php?release=2014-01

[4]https://github.com/moses-smt/mosesdecoder/blob/master/scripts/tokenizer/lowercase.perl

[5]https://github.com/rsennrich/subword-nmt

[6]http://nlp.stanford.edu/projects/nmt

[7]https://github.com/moses-smt/mosesdecoder/blob/master/scripts/generic/multi-bleu.perl

[8]For transformer_base/transformer_big configuration, our reproduce score is 27.35/28.24 while the score in the paper is 27.30/28.40.

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
