[Supplementary Material]

# A    Case Study

Following Section 5.3, we provide more visualizations and analyses on attention weights. As shown in Figure 1, 2, 3, in this case, our layer-wise coordination model translates the source sentence completely, while Transformer misses the words "and what" (translated from "und was"). From Figure 3, we can easily find that our model correctly attends on "und" when generating "and". In contrast, Transformer attends on "wie" and "und" simultaneously, resulting in word missing.

Figure 1: Source to target attention in Transformer.

Figure 2: Target self-attention in Transformer.

Figure 3: Mixed attention in our model.

In another case as shown in Figure 4, 5, 6, Transformer misses the subject of the sentence. We can observe that Transformer gives more attention on the word "reinigt" (which means "clean" in English) than "es" (which means "it" in English) from Figure 4. It should be pointed out that although "es" is also be noted, the introducing of attention noise may prevent the generation of "it", leading to a incomplete translation. In Figure 6, "es" and "," are the most important parts our model attends on, this reflects that our layer-wise coordination can pay attention to both source and target sentences correctly.

Figure 4: Source to target attention in Transformer.

Figure 5: Target self-attention in Transformer.

Figure 6: Mixed attention in our model.

## B   The Implementation of Context Gates on Transformer

The context gates proposed by [27] is originally implemented in RNN, where the basic idea is using gates to control the amount of information of the attention results (contexts) from source tokens and from previous target tokens when generating the current target token. In this paper we implement in Transformer for fair comparison. Denote $C_i^{tgt}$ as the results of the target self-attention (target contexts) and $C_i^{src}$ as the results of source-target attention (source contexts) on position $i$. The context gating mechanism is implemented as:

$$Z_i = \sigma(W_{src}C_i^{src} + W_{tgt}C_i^{tgt}), \tag{1}$$

where $\sigma(.)$ is a logistic sigmoid function, $W_{src}$ and $W_{tgt} \in \mathbb{R}^{d_{\text{model}}*d_{\text{model}}}$, and $d_{\text{model}}$ is the dimension of the hidden size.

Transformer originally leverages a residual connection to add $C_i^{tgt}$ and $C_i^{src}$ and feed the addition results to the next sublayer. Here we apply the context gates on this addition to get the final result:

$$C_i = Z_iC_i^{src} + (1 - Z_i)C_i^{tgt}. \tag{2}$$

We keep other components of the model the same as the Transformer model.

## C   Discussion and Visualization of Mixed Attention

Intuitively, the generation of content words (e.g., the named entity) depend more on source contexts while the function words (e.g., of, on, the) depend more on target contexts. However, when generating the target word, the state-of-the-art NMT model [30] extracts the source and target contexts with two separate attention mechanisms. Therefore, the total amount of information extracted from the source and target contexts are equal and there is no preference on the two contexts. Our mixed attention mechanism will facilitate the model to learn to extract the information from source and target more selectively.

Here we show three situations where our mixed attention works well. In situation 1 as shown in Figure 7, the generation of current token mostly depends on the previous target tokens. In situation 2 as shown in Figure 8, the generation of current token mostly depends on the source tokens. In situation 3 as shown in Figure 9, the generation of current token depends on both the source tokens and target tokens. In the first two situations, the current token just needs more information from the target (in situation 1) and source (in situation 2) contexts. However, the conventional attention mechanism captures both the source and target context through two separate attention mechanisms, and thus it may bring unnecessary information. Our mixed attention can selectively choose which contexts to depend on, which is more focused and accurate. In the third situation where both source and target contexts are needed, our mix attention also captures the information well, without only attending to the source part or the target part of the sentence.

Figure 7: A case of mixed attention on De-En translation, where generating the target token "of" mostly depends on the previous target token "lot", which together formulate the phrase "a lot of".

Figure 8: A case of mixed attention on De-En translation, where generating the target token "second" mostly depends on the source token "zweite" which means "second" in English.

Figure 9: A case of mixed attention on De-En translation, where generating the target token "indians" depends on both the source token "indianer" which means "indians" in English and the previous target token "the" simultaneously.