[Reviews · NeurIPS 2018]

Reviewer 1



Thanks for the author feedback! ---- The paper proposes a modification to how attention is performed in sequence-to-sequence models. Instead of attending at the last layer of the encoder, each layer in the decoder attends to the corresponding layer in the encoder. This modification allows the models to cover more source words compared to vanilla models. The paper presents a good comparison to previous methods. The visualization provide useful information on what the model has learned and the ablation studies are also much appreciated. It is very interesting that the layer-wise attention helps them get better performance as they increase the number of layers compared to baseline model.

Reviewer 2



Update after Author Feedback: Thanks for all the extra info about speed and alternative encoder-decoders. Original Review: This work builds directly off of Transformer networks. They make two contributions to that kind of architecture. The first is to suggest running the encoder and decoder stacks layer by layer instead of running the encoder stack and passing information to the decoder stack. The second is to actually tie the weights of the encoder and decoder. +: Shows that sharing the weights between the encoder and decoder of a transformer network is promising.This is surprising and worth knowing. Running a decoder layer right after its corresponding encoder layer processes (rather than running the next encoder layer) is also an interesting augmentation to Transformer networks. Again surprising that this contributes to better performance. If this has been done in the past, I certainly haven’t seen it in recent work. Adding the learned source and target tokens is an interesting method for ensuring that the model knows which language the embeddings are in. Strong improvements on NMT over the baseline Transformer network. -: Giving all layers of the decoder access to corresponding layers in the encoder does not seem novel, at least in the way it is phrased at various points throughout the paper. Though the original Transformer paper (Vaswani et al 2017) did in fact use only the top level outputs of their encoder, Figure 1 of (Vaswani et al 2017) actually more closely depicts the corresponding access to encoder layers. Perhaps because of that depiction, some of the follow up work, for example Gu et al. 2017, did actually use layer wise attention without considering that a major contribution of their work. This work only shows layer wise coordination for the Transformer, not for other kinds of encoder-decoders, so its not clear that this coordination is more generally useful. To make the broader claim, I think it is necessary to show this idea is empirically fruitful for at least one of CNNs or LSTMs as well. This is mentioned as future work, but as is, this comes off more as a tweak of the Transformer than evidence of a more generally applicable idea. You have to have 14 layers instead of 6 by doing it this way. How does this increase in layers combined with layer-wise processing affect efficiency at train/inference time? This paper did indeed shift the way that I would try doing seq2seq tasks in the future, and the results on NMT were strong. On first read, it seemed like an incremental change to Transformers, but after looking more closely, I do think it is worth it for the community to try out this different perspective on encoder-decoder architectures. Overall 7, confidence 4. The code is said to be open sourced after reviewing, so I'm very confident this will be reproducible.

Reviewer 3



This work proposes a layer-wise coordination for encoder and decoder networks of sequence-to-sequence model by jointly computing attention to both of encoder and decoder representation of each layer. In a conventional transfomer model, each decoder layer computes attention over the last layer of encoder layers, and self-attention to the decoder's history and attention over encoder representation are computed separately as different layers. In this work: - Encoder and decoder share the same number of layers. - Each decoder layer computes attention to its corresponding encoder layer, not the last encoder layer. - Attention does not differentiate encoder or decoder, but they are simply mixed so that attention layer could focus either on particular position at encoder or decoder. Experimental results indicate significant gains over strong baselines under multiple language pairs. It is an interesting work on improving the strong transformer baselines, and experiments look solid. I think larger gains might come from more layers since the proposed model does not have to differentiate whether it is performing self attention and/or encoder attention. However I have a couple of concerns to this work. - It is still not clear to me why mixing encoder and decoder for computing attention is better though attention visualization could be a clue. It would be good to analyze whether particular layers might be serving only for self-attention and/or encoder-attention. - I could not find any notions of multi-head attention, though it is one of the features in transformer which implicitly performs projection from encoder attention to decoder representation. Do you use multi-head attention and if so, how many heads employed in your experiments? If no multi-head attention is employed in this work, it is a bit of mystery in that encoder representation will not be projected into decoder representation. As indicated in the response, please mention it. Other minor comment: - It has an implicit assumption that source/target token ids must be shared given the embedding parameters are shared. If possible, I'd like to see the effect of using the shared embedding for the original transformer model.